# What are the important morbidities associated with paediatric cardiac surgery? A mixed methods study

Katherine L Brown,[1] Christina Pagel,[2] Deborah Ridout,[3] Jo Wray,[1] David Anderson,[4] David J Barron,[5] Jane Cassidy,[5] Peter Davis,[6] Emma Hudson,[7] Alison Jones,[5] Andrew Mclean,[8] Stephen Morris,[9] Warren Rodrigues,[10] Karen Sheehan,[11] Serban Stoica,[12] Shane M Tibby,[13] Thomas Witter,[4] Victor T Tsang,[1] Cardiac Impact Study Group

For numbered affiliations see end of article.

**Correspondence to**
Dr Katherine L Brown;
katherine.brown@gosh.nhs.uk

## ABSTRACT

**Objectives** Given the current excellent early mortality rates for paediatric cardiac surgery, stakeholders believe that this important safety outcome should be supplemented by a wider range of measures. Our objectives were to prospectively measure the incidence of morbidities following paediatric cardiac surgery and to evaluate their clinical and health-economic impact over 6 months.

**Design** The design was a prospective, multicentre, multidisciplinary mixed methods study.

**Setting** The setting was 5 of the 10 paediatric cardiac surgery centres in the UK with 21 months recruitment.

**Participants** Included were 3090 paediatric cardiac surgeries, of which 666 patients were recruited to an impact substudy.

**Results** Families and clinicians prioritised: Acute neurological event, unplanned re-intervention, feeding problems, renal replacement therapy, major adverse events, extracorporeal life support, necrotising enterocolitis, postsurgical infection and prolonged pleural effusion or chylothorax.

Among 3090 consecutive surgeries, there were 675 (21.8%) with at least one of these morbidities. Independent risk factors for morbidity included neonatal age, complex heart disease and prolonged cardiopulmonary bypass (p<0.001). Among patients with morbidity, 6-month survival was 88.2% (95% CI 85.4 to 90.6) compared with 99.3% (95% CI 98.9 to 99.6) with none of the morbidities (p<0.001). The impact substudy in 340 children with morbidity and 326 control children with no morbidity indicated that morbidity-related impairment in quality of life improved between 6 weeks and 6 months. When compared with children with no morbidities, those with morbidity experienced a median of 13 (95% CI 10.2 to 15.8, p<0.001) fewer days at home by 6 months, and an adjusted incremental cost of £21 292 (95% CI £17 694 to £32 423, p<0.001).

**Conclusions** Evaluation of postoperative morbidity is more complicated than measuring early mortality. However, tracking morbidity after paediatric cardiac surgery over 6 months offers stakeholders important data that are of value to parents and will be useful in driving future quality improvement.

## Strengths and limitations of this study

► Our study is unique, given that the morbidity measures selected for study reflect the viewpoints of both lay people and clinicians, whereas previous work has tended to focus on clinical metrics only.

► A limitation of our study is that although we were able to identify broad risk factors for morbidity after paediatric cardiac surgery, a larger dataset will be needed to generate more reliable risk adjustment methods.

► A strength of our study was that it was multicentre, prospective and followed children up for 6 months after their operation, which is a longer time horizon than the usual 30-day period.

## BACKGROUND

Over 5000 paediatric cardiac surgery procedures are performed in the UK each year and early survival has improved to over 98% since comprehensive national audit commenced in 2000.[1][2] Most stakeholders including clinicians, commissioners and users believe that while these early survival rates remain an important safety measure, it has become imperative to explore a broader range of measures for outcome in this complex field of practice. While there has been considerable research related to measuring, understanding and reducing perioperative mortality for paediatric cardiac surgery,[3–5] much less attention has been focused on surgical morbidities.

Morbidity is defined as a state of being unhealthy, or of experiencing an aspect of health that is 'generally bad for you'. In this project, by 'morbidity' we mean a defined aspect of ill health associated with a specific operation. In our study, we aimed to identify which morbidities present the greatest burden on patients and health services following paediatric cardiac surgery. Views

may differ between professionals and non-professionals as to what the term morbidity exactly refers to and which surgical morbidities are most important.[6] Therefore, we set out to combine patients' and carers' perspectives with those of professional groups in defining a prioritised list of morbidities for prospective evaluation.

Over 4 years, we:
1. Identified and defined nine morbidities following paediatric heart surgery, taking into account views from patients, carers, psychologists, nurses and clinicians, which together captured important aspects of the clinical and health-economic burden;
2. Measured incidence of the defined morbidities in the UK patient population and in subgroups defined by case complexity;
3. Evaluated the impact of defined morbidities on quality of life and estimated their clinical and health economic burden.

## METHODS
### Design
The design was a prospective, multicentre, multidisciplinary mixed methods study. Selection of morbidities was based on consensus methods. Morbidity incidence was evaluated with a prospective cross-sectional study. Measurement of morbidity impact entailed a prospective, case-matched, longitudinal study.

### Setting
Five UK paediatric cardiac surgery centres representing a range of programme sizes, which care for half of all patients nationally.

### Participants
The participants were children aged under 17 years with congenital heart disease (CHD).

### Consent
Parents and children participating in focus groups and in the prospective study of morbidity impact provided written informed consent.

### Which morbidities?
Between January 2014 and September 2015, we reviewed existing literature, ran three family focus groups and undertook a family online discussion forum moderated by the Children's Heart Federation (a user group). Transcripts were thematically analysed and the resultant themes, together with the literature, helped to inform a long list of candidate surgical morbidities. A multidisciplinary group, with patient and carer representation, then ranked and selected a list of nine key morbidities using the nominal group technique and secret voting.[7] This 'selection panel'[7] was informed in turn by clinical views on definitions and feasibility of routine monitoring for each candidate morbidity provided by an independent 'definition panel'[8] as reported in previous publications.

### Incidence of morbidity
Between October 2015 and June 2017, we prospectively measured instances of morbidity within all consecutive surgical admissions across the study sites. Online supplementary appendix 1[8] provides the details of the criteria followed to define each individual morbidity, including the timeframe for diagnosis. Morbidities were attributed to the immediately preceding cardiac surgery and defined within the same hospitalisation. The only exceptions were unplanned re-operation, which was defined as an unanticipated procedure within 30 days and mediastinitis, which could be identified postdischarge by the operating surgeon.[8] Data were regularly checked for completeness, clinical congruence and accuracy and, a 3-month sample of data was independently validated against the national audit data.

As for the UK audit of 30-day mortality,[9] all procedures >30 days apart on the same patient were included in the morbidity analysis. As a secondary outcome, we checked each patient's survival status at 6 months after first appearance in the dataset, based on the individual centres National Congenital Heart Diseases Audit (NCHDA)[2] data and local hospital records.

### Impact of morbidity
#### Recruitment
Recruitment to the impact study ran between October 2015 and June 2017 for all but one site, where it stopped after 6 months due to resource constraints. We attempted to recruit all patients with morbidity from the wider population, when at least one parent spoke reasonable English, and the family were resident in the UK. When feasible, for each morbidity case, we recruited a patient with no morbidity, matched on treating centre, age and univentricular status. The recruitment strategy led to the recruitment of a sample that was evenly balanced between patients with morbidity and those with no morbidity, however we were not able to find a match for every morbidity patient. In order to account for the widest possible spectrum of important outcomes, we included the available data on children who subsequently died.

#### Measures of impact
We evaluated the impact of morbidities over the 6 months following surgery, based on the outcomes discussed below.

#### *Quality of life and psychological burden on children and parents*
We used age-specific measures at 6 weeks and 6 months following surgery. These were the PedsQL subscales (physical and psychosocial) and total scores. These scores range from 0 and 100, a higher score indicating better quality of life. PedsQL scores in a normal healthy population vary by age; expected scores for infants 0–12 months (which encompasses the median age for our cohort) are mean physical 84.98 (SD 9.45), mean psychosocial 80.47 (SD 12.64) and mean total 82.47 (SD 9.95).[10–12]

For the parents, we used the Patient Health Questionnaire-4 (PHQ-4), which comprises four questions,

measuring anxiety and depression in adults. Individual items are scored from 0 to 3. Scores ≥3 for the first two questions suggest anxiety, and scores ≥3 for last two questions suggest depression.[13] In the normal population, 4.8% have scores suggestive of anxiety and 6.6% have PHQ-4 scores suggestive of depression.[14]

### Days at home over 6 months as an additional measure of disruption to family life

Length of inpatient stay following the index procedure was defined as the number of days between the operation and the date of discharge from the tertiary centre. Hospital stays that exceeded the 6-month follow-up period were censored after 183 days. All subsequent hospital admissions were captured and the total days spent in hospital including tertiary and secondary care admissions was subtracted from 183 days to generate a value for days at home by 6 months. Children who died in hospital were assigned a value of zero for this outcome.

### Costs of the index hospitalisation

We calculated the cost of the inpatient stay in the tertiary centre following the index surgical procedure as a key indicator of the economic burden to the hospital provider (measured in 2016/17 UK£).

Costs were calculated as the summation of days in intensive care unit (ICU), days on the ward and all operating procedures within the defined period. For ICU stays, data were recorded on the level of care the patient received each day (enhanced care, high dependency, high dependency advanced, ICU basic, ICU basic enhanced, ICU advanced, ICU advanced enhanced and extracorporeal life support (ECLS)).[15] Unit costs of ICU stays were daily costs for each level of care (ranging from £870 to £5440) applied to the number of days spent receiving that level of care. Unit costs per day for ward stays were obtained from the highest recruiting centre and varied according to the age of the patient (<2 years, £904; ≥2 years, £680). Unit costs for surgical procedures were costs per minute supplied by study sites; these varied by site and so we used the value for the site that recruited the largest number of patients to this study (£13.39 per min), which was similar to the mean value across all sites.

### Statistical and health economic analyses
#### Patient sample size

We anticipated that between 3000 and 3300 surgical patients would be admitted across the five sites during the study period.[2] We assumed that a clinically relevant difference in quality of life between matched pairs corresponded to a mean difference in quality of life score of at least 0.5 SD.[16] To detect such a difference for PedsQL responses at 5% significance with 80% power requires a minimum of 32 matched pairs. Allowing for a 10% loss to follow-up rate, we aimed to recruit 36 matched pairs for each patient with morbidity, giving 80% power to detect a significant effect for any morbidity with a prevalence of at least 1.5%.

### Analysis of risk factors for morbidity

Based on the whole sample of cardiac surgeries across the five sites over the study period, we explored risk factors for occurrence of morbidity. Clinical risk factor groups were derived from the finer diagnostic coding, based on previous peer-reviewed research by our group.[9 17] We used multilevel logistic regression analysis to explore the role of preoperative, patient-level case mix factors on the occurrence of any morbidity versus no morbidity, accounting for multiple procedures within patients. First, a univariate model to predict risk of any morbidity versus none was fitted for each risk factor. The estimated ORs are presented along with 95% CIs. Then all factors significant on univariate analysis (p<0.1) were considered in a multivariable model. We used multiple imputation by chained equations to account for missing data. The imputation model included all risk factors considered in the univariate analysis, which we assumed included all predictors of missingness. We indicate missing data in our results. The multivariable models were derived by fitting a regression model for all significant predictors and estimates were combined using Rubin's rules.[18]

### Secondary outcome

The secondary outcome of survival at 6 months was calculated for all patients who had cardiac surgery at the sites within the study period. Unadjusted 6-month mortality rates were compared between morbidity groups using logistic regression.

### Analysis of the impact of morbidity

In the patients recruited to the impact study, we analysed the 'impact outcomes' over 6 months postsurgery.

We used mixed effects regression models for PedsQL/PHQ-4 results to compare the impact of morbidity (any vs none) on outcome. All models were adjusted for clustering within matched pairs and significant covariates associated with incidence of morbidity. We used multiple imputation, by chained equations to account for missing data for those patients known to be alive at 6 weeks and 6 months, respectively.

We used quantile regression to estimate the effects of individual morbidities in terms of differences in median days at home by 6 months, since the measure was negatively skewed. All models were adjusted for clustering within matched pairs and significant covariates associated with incidence of morbidity. Given the high level of data completeness, imputation was not used.

The inclusion of hospital site made negligible difference to either of these analyses, and not included in the statistical models.

We modelled the relationship between morbidities and cost of index hospitalisation, adjusted for significant covariates associated with hospital costs. Costs were skewed and to account for this we used a generalised linear model with gamma family and log link, which has been recommended for modelling positively skewed data.[19] Missing outcome values were imputed using an iterative Markov chain Monte

Carlo procedure based on multivariate normal regression, generating 20 imputed datasets. The hospital site was included as a significant covariate in the health economic model.

All analyses were performed in Stata V.14.[20]

## Patient and public involvement

The patient or user perspective lies at the core of the study methodology. Specifically, a key goal of the study was to consider the views of patients and parents when measures of morbidity are selected for future audit and benchmarking, in particular since emphasis may potentially differ between professionals and parents/patients.

Patient and family representatives from the Children's Heart Federation (CHF) were involved in aspects of the study design including design and running of the focus groups and the online discussion forum.

Representatives of the CHF participated in the facilitated nominal group meetings to select morbidities for inclusion in the incidence and impact studies.

A member of the CHF and a parent representative sat on the project steering group.

## RESULTS
### Which morbidities?

Our final selected[7] and defined[8] list of included morbidities were: acute neurological event, unplanned re-intervention, feeding problems, renal replacement therapy, major adverse events, ECLS, necrotising enterocolitis, postsurgical infection and prolonged pleural effusion or chylothorax.

We treated children with more than one of these events as a distinct group referred to as 'multimorbidity'. Recognising ECLS as a very severe event, which nearly always occurs with other morbidities, we treated ECLS as stand-alone morbidity (and always excluded from the multiple morbidity group).[21–25]

There was some divergence between the views of clinicians and families about the fundamental issue of what the important morbidities linked to paediatric cardiac surgery are. Health professionals from tertiary centres tended to prioritise clearly clinical issues related to the heart (eg, use of ECLS and re-operation), whereas parents placed greater emphasis on holistic outcomes for their child (eg, feeding difficulties and child developmental problems).[7]

### 2.Incidence of morbidity
#### Descriptive results

We collected data on 3248 cardiac procedures and analysed 3090, after excluding 99 cardiac re-operations (a morbidity outcome) and 59 procedures that did not meet inclusion criteria.

The incidence of individual morbidities in isolation and overall for each is shown in figure 1.

#### Significant risk factors for morbidity

As shown in table 1, the most important risk factor for occurrence of 'any morbidity' was young age, particularly neonates, who were approximately five times as likely to experience a morbidity compared with children aged over 1 year. Children with more complex heart disease or children whose operation included a

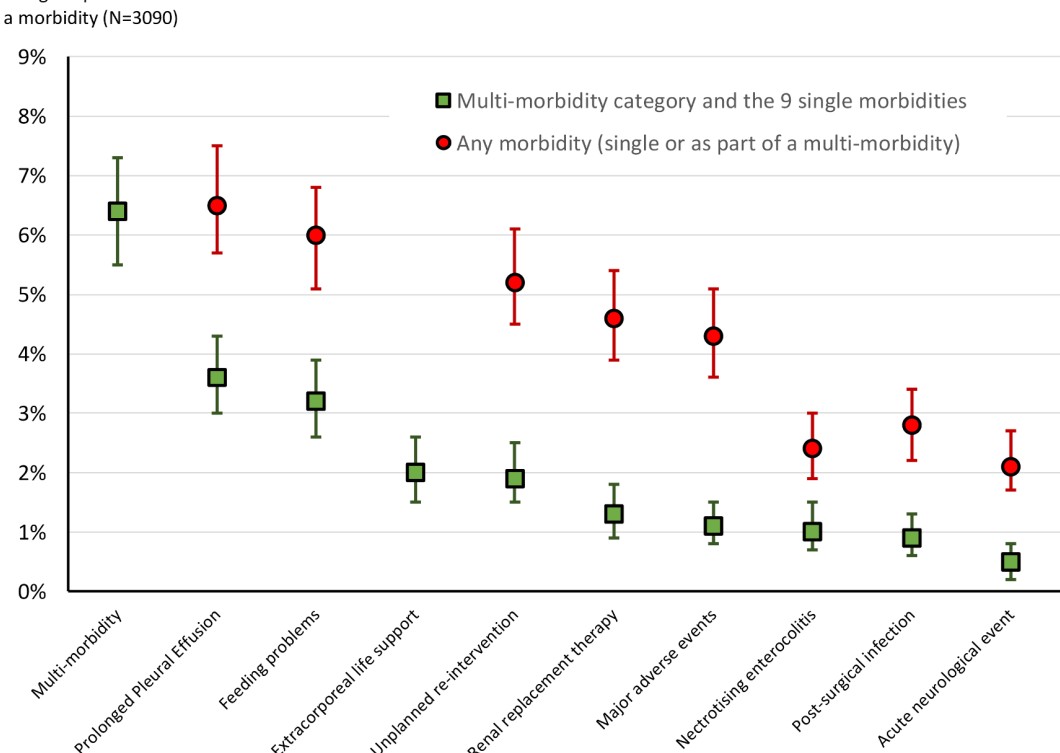

**Figure 1** Incidence of morbidities. The figure shows the distinction between occurrence as a single morbidity and occurrence in combination with other morbidities, with 95% CIs. **Source**: Reproduce from Brown et al[35] 2019.

**Table 1** Summary of risk factors by any morbidity outcome

| | None of the morbidities (N=2415) Number (%) or (IQR where stated) | Any morbidity (N=675) Number (%) or (IQR where stated) | Univariate OR for 'any morbidity' (95% CI), p value | Multivariable OR for 'any morbidity' (95% CI), p value |
|---|---|---|---|---|
| Male | 1299 (53.8) | 372 (55.1) | 1.05 (0.89 to 1.25), 0.54 | – |
| Median age (days) (IQR) | 286 (105, 1582) | 102 (10, 331) | | |
| Child | 1111 (46.0) | 160 (23.7) | | |
| Infant | 1023 (42.4) | 268 (39.7) | 1.82 (1.47 to 2.25), <0.001 | 1.61 (1.26 to 2.05), <0.001 |
| Neonate | 281 (11.6) | 247 (36.6) | 6.10 (4.81 to 7.75), <0.001 | 5.26 (3.90 to 7.09), <0.001 |
| Median weight (kg) (IQR) | 7.7 (4.7, 16.2) | 4.6 (3.2, 8.0) | | |
| Low weight <2 SD below mean | 714 (31.5) | 234 (36.8) | 1.27 (1.05 to 1.52), 0.01 | 1.21 (0.97 to 1.51), 0.09 |
| Cardiac diagnosis group | | | | |
| E—Least complex (reference) | 1002 (41.5) | 123 (18.2) | | |
| D | 796 (33.0) | 227 (33.6) | 5.13 (3.79 to 6.93), <0.001 | 2.02 (1.58 to 2.60), <0.001 |
| C | 215 (8.9) | 109 (16.2) | 3.83 (2.85 to 5.13), <0.001 | 1.44 (1.00 to 2.07), 0.05 |
| B | 232 (9.6) | 109 (16.2) | 4.13 (3.07 to 5.55), <0.001 | 2.62 (1.85 to 3.71), <0.001 |
| A—Most complex | 170 (7.0) | 107 (15.8) | 2.32 (1.83 to 2.94), <0.001 | 2.14 (1.41 to 3.24), <0.001 |
| Functionally univentricular heart | 255 (10.6) | 159 (23.6) | 2.61 (2.11 to 3.23), <0.001 | 1.55 (1.07 to 2.24), 0.02 |
| Acquired comorbidity | 337 (14.0) | 119 (17.6) | 1.32 (1.05 to 1.66), 0.02 | 1.33 (1.03 to 1.71), 0.03 |
| Congenital comorbidity | 537 (22.2) | 178 (26.4) | 1.25 (1.03 to 1.52), 0.03 | 1.28 (1.02 to 1.59), 0.03 |
| Severity of illness indicators | 222 (9.2) | 152 (22.5) | 2.87 (2.30 to 3.58), <0.001 | 1.52 (1.16 to 2.00), <0.01 |
| Premature <37 weeks gestation | 231 (9.6) | 73 (10.8) | 1.15 (0.87 to 1.51), 0.33 | – |
| Down syndrome | 214 (8.9) | 63 (9.3) | 1.06 (0.79 to 1.43), 0.71 | – |
| Additional cardiac risk | 165 (6.8) | 65 (9.6) | 1.45 (1.09 to 1.94), 0.01 | 1.39 (0.99 to 1.94), 0.05 |
| Surgical procedure type | | | | |
| Reparative/corrective (reference) | 1391 (57.6) | 332 (49.2) | | |
| Palliative/staged | 331 (13.7) | 179 (26.5) | 2.27 (1.82, to 2.82), <0.001 | 1.65 (1.14 to 2.38), <0.01 |
| Ungrouped | 693 (28.7) | 164 (24.3) | 0.99 (0.81 to 1.22), 0.94 | 1.04 (0.82 to 1.31), 0.75 |
| Median bypass time (min) (IQR) | 72 (42, 110) | 110 (62, 156) | | |
| No bypass (reference) | 390 (16.2) | 103 (15.3) | | |
| Up to 90 min | 1148 (47.5) | 150 (22.2) | 0.48 (0.35 to 0.65), <0.001 | 0.78 (0.57 to 1.09), 0.14 |
| >90 min | 877 (36.3) | 422 (62.5) | 1.76 (1.32 to 2.34), <0.001 | 2.28 (1.67 to 3.12), <0.001 |

The weight was missing in 87 (2.8%) procedures and weight for age was set to missing for an additional 99 (3.2%) patients in whom the value was >5 SD from the normative mean. Other data were complete. Values include imputed data.

Cardiac diagnosis group (A) hypoplastic left heart syndrome, truncus arteriosus, pulmonary atresia intact septum, (B) functionally univentricular heart, pulmonary atresia ventricular septal defect, (C) transposition of the great arteries all types, interrupted aortic arch, totally anomalous pulmonary venous connection, (D) patent ductus arteriosus, tricuspid valve anomalies, acquired heart disease, complete atrioventricular septal defect, (E) tetralogy of Fallot, mitral valve anomalies, isolated aortic stenosis, aortic regurgitation, aortic arch obstruction, subaortic obstruction, ventricular septal defect, atrial septal defect.

The results of the multiple logistic regression model for occurrence of 'any morbidity' are expressed as OR for 'any morbidity' by the stated factor, adjusted for age band, low weight, cardiac diagnostic category, functionally univentricular heart, acquired comorbidity, congenital comorbidity, severity of illness indicators, additional cardiac risk factors, specific procedure group, bypass time category.

**Source**: Reproduce from Brown et al[35] 2019

bypass time longer than 90 min were almost twice as likely to experience morbidity. Other important independent risk factors were: palliative or staged procedures, the presence of functionally univentricular heart and preprocedure severity of illness factors (eg, preprocedure mechanical ventilation or shock).

### Secondary outcomes

Patient survival was highest and patient hospital stay was shortest for patients with none of the morbidities (table 2). The 6-month survival was significantly lower for patients with a single morbidity, for those with ECLS and multiple morbidities, compared with patients with none of the morbidities (p<0.001 for all).

### Impact of morbidity
#### Patient sample

We recruited 60% of eligible families, amounting to 666 children of whom 340 (51%) had at least one morbidity. Of these, 19 died within 30 days and 39 died within 6 months. We pair matched 558 patients and 108

**Table 2** Secondary outcomes of hospital stay and survival at 6 months by morbidity

| Morbidity type | Number (%) with morbidity | Median length of postoperative hospital stay in days (IQR) | 6-month survival Number, % (95% CI) |
|---|---|---|---|
| No morbidity | 2415 (78.2%) | 8 days (5, 13) | 2202/2217 99.3% (98.9 to 99.6) |
| Any morbidity | 675 (21.8%) | 24 days | 675 (21.8%) |
| Acute neurological event | 14 (0.5) | 19 days (12, 39) | 12/13 92.3% (64.0 to 99.8) |
| Unplanned re-intervention | 59 (1.9%) | 22 days (14, 33) | 50/54 92.6% (82.1 to 97.9) |
| Feeding problems | 99 (3.2%) | 20.5 days (12, 36) | 90/91 98.9% (94 to 100) |
| Renal support | 40 (1.3%) | 17 days (14, 26) | 37/39 94.9% (82.7 to 99.4) |
| Major adverse event | 34 (1.1) | 16.5 days (8, 25) | 28/33 84.9% (68.1 to 94.9) |
| Necrotising enterocolitis | 32 (1.0%) | 24.5 days (18.5, 49.5) | 28/30 93.3% (77.9 to 99.2) |
| Postsurgical infection | 27 (0.9%) | 20.5 days (11, 28) | 25/25 100% (86.3 to 100) |
| Prolonged pleural effusion | 111 (3.6%) | 20 days (14, 28) | 95/96 99.0% (94.3 to 100) |
| Multimorbidity | 197 (6.4) | 35 days (22, 56) | 158/190 83.2% (77.1 to 88.2) |
| Extracorporeal life support | 62 (2.0%) | 43 days (20, 84) | 31/57 54.4% (40.7 to 67.6) |

The number (%) of morbidities was based on individual surgical episode, as was the length of stay, whereas survival was calculated per patient. Data are presented based on morbidity in isolation, multimorbidity and extracorporeal life support. Length of stay data were missing for 9 patients and life status was missing in 16 patients who were not included.

patients unmatched of whom 61 had morbidity; all of the patients were included in the statistical analysis.

### Quality of life and anxiety—depression outcomes

For the 6 weeks data, a comparison between patients with missing versus non-missing data indicated the only difference was in the proportions with severity of illness factors and additional cardiac risk factors. Importantly, there was no difference in the proportion of patients with or without morbidity. At the 6 months mark, there was no difference in any baseline factor or morbidity occurrence.

All PedsQL scores were lower than in a healthy population (see 'Methods' section[10–12] and table 3). In terms of our study objective to assess the impact of morbidity, the total PedsQL scores were significantly reduced for children with any morbidity at 6 weeks in comparison to children with none of the morbidities (case mix adjusted score reduction of –5.2 (95% CI –8.3 to –2.2, p<0.01). On a positive note, this difference narrowed to small non-significant reduction in total score by 6 months. Physical scores were affected more than psychosocial scores, and were still significantly impaired at 6 months.

All parents experienced higher rates of both anxiety and depression than a healthy population (see

'Methods' section and table 3). The parents of children with morbidity were around 57% more likely to experience anxiety; and around 77% more likely to experience depression, at 6 weeks postoperation than parents of children without a morbidity. PHQ-4 scores had improved by 6 months, and although there remained a higher chance of both anxiety and depression with morbidity, the difference narrowed and was not significant.

### Days at home by 6 months outcome

Children with none of the morbidities had a median of 174 days at home (IQR 169, 176) out of a possible 183 days in 6 months. The presence of any morbidity reduced the median days at home by 13 days over 6 months (95% CI 10.2 to 15.8, p<0.001). When individual morbidities were each considered in turn, the difference in the case mix adjusted median days at home was significantly lower at 6 months for all of them apart from 'renal support' (p<0.05).

### Health economic outcome

The costs of the index hospitalisation were available for 613 patients (8% missing). The mean additional cost linked to having any of the selected morbidities

**Table 3** Quality of life and parental anxiety/depression based on whether or not a morbidity was present

| The impact outcome type and time point of measurement | Patients with none of the morbidities | Patients with any morbidity | Differences in outcome based on presence of any morbidity |
|---|---|---|---|
| | *Score mean (SD)* | *Score mean (SD)* | *Adjusted difference in mean score with morbidity (95% CI), p value* |
| Physical QOL score at 6 weeks | 79.0 (16.2) | 69.1 (21.8) | −8.3 (−11.8 to −4.9), <0.001 |
| Psychosocial QOL score at 6 weeks | 79.4 (14.7) | 75.2 (19.0) | −2.7 (−5.9 to 0.5), 0.08 |
| Total QOL score at 6 weeks | 79.3 (13.8) | 72.1 (19.1) | −5.2 (−8.3 to −2.2), <0.01 |
| Physical QOL score at 6 months | 82.3 (16.6) | 76.6 (18.6) | −4.2 (-7.6 to −0.8), 0.02 |
| Psychosocial QOL score at 6 months | 78.0 (14.7) | 76.4 (15.4) | −0.9 (-3.5 to 1.8), 0.5 |
| Total QOL score at 6 months | 79.8 (13.9) | 76.6 (15.0) | −2.3 (-4.9 to 0.3), 0.08 |
| | *Number (%) with attribute* | *Number (%) with attribute* | *Adjusted OR for outcome by morbidity (95% CI), p value* |
| Parental anxiety at 6 weeks | 55 (23.7) | 92 (36.8) | 1.57 (1.04 to 2.35), 0.03 |
| Parental depression at 6 weeks | 26 (11.2) | 55 (22.1) | 1.77 (1.10 to 2.86), 0.02 |
| Parental anxiety at 6 months | 23 (11.5) | 34 (17.4) | 1.37 (0.77 to 2.43), 0.28 |
| Parental depression at 6 months | 15 (7.5) | 23 (11.9) | 1.45 (0.79 to 2.68), 0.23 |

The differences in each outcome are adjusted for age band, low weight, cardiac diagnostic category, functionally univentricular heart, acquired comorbidity, congenital comorbidity, severity of illness indicators, additional cardiac risk factors, specific procedure group, bypass time category.
Values include imputed data.
Quality of life was derived from PedsQL scores for 478 children at 6 weeks (26% missing) and 403 children at 6 months (36% missing).
Parental anxiety and depression was derived from PHQ-4 responses of 481 parents at 6 weeks (26% missing) and 394 parents at 6 months (37% missing).
PHQ-4, Patient Health Questionnaire-4; QOL, quality of life.

was £21 292 (95% CI £17 694 to £32 423, p<0.001). In table 4, we present the adjusted marginal effects, which are the adjusted incremental costs linked to the stated morbidity, in comparison to the adjusted cost linked to patients with none of the selected morbidities. The greatest adjusted cost difference was for ECLS, where the adjusted mean difference in costs was £62 452

followed by multimorbidity, where the adjusted mean difference was £33 147 (p<0.01 for both). Significantly increased (p<0.05) costs were also found for unplanned re-intervention, feeding problems, renal support and prolonged pleural effusion. The remaining morbidities, representing the lowest number of patients per category, showed non-significant differences.

**Table 4** Cost of hospital stay following index surgical procedure by morbidity

| Morbidity | Coefficient | Marginal effect (£) (95% CI), p value |
|---|---|---|
| Any morbidity (n=340) | 0.71 | 21 292 (17 694 to 32 423), <0.001 |
| Acute neurological event (n=6) | 0.50 | 13 911 (−3923 to 50 059), 0.16 |
| Unplanned re-intervention (n=26) | 0.62 | 18 330 (7475 to 33 282), <0.01 |
| Feeding morbidity (n=45) | 0.55 | 15 541 (7200 to 26 326), <0.01 |
| Renal support (n=24) | 0.41 | 10 798 (1846 to 23 215), 0.01 |
| Major adverse event (n=22) | 0.30 | 7555 (−1131 to 19 984), 0.10 |
| Necrotising enterocolitis (n=11) | 0.37 | 9547 (−2173 to 28 454), 0.13 |
| Postsurgical infection (n=11) | 0.27 | 6466 (−3947 to 23 130), 0.17 |
| Prolonged pleural effusion (n=50) | 0.33 | 8177 (1444 to 16 904), 0.01 |
| Extracorporeal life support (n=27) | 1.37 | 62 452 (39 546 to 93 983), <0.01 |
| Multimorbidity (n=118) | 0.94 | 33 147 (23 669 to 44 624), <0.01 |

The marginal effects are adjusted for age band; weight-by-age z-score; congenital morbidity, severity of illness indicator, Down syndrome; cardiac diagnosis category; cardiac procedure category; whether or not the patient died within 6 months and study site.
Figures in the 'Marginal effect' column are the mean difference in costs between each morbidity category and the category of no selected morbidities present, conditional on the covariates. Costs are in 2016/17 UK£. Values include imputed data.

## DISCUSSION

### Overview of findings

This paper presents the results of the first prospective multicentre study of morbidity after paediatric cardiac surgery morbidity that measured outcome 6 months after operation. The results are novel and indicate that evaluation of morbidity adds considerably to the current main metric of 30-day or in-hospital survival.

We note the excellent outcomes in children who had none of the morbidities: their survival at 6 months post-operation was 99.3% and their median length of stay was 8 days, suggesting that the morbidities we selected do capture most of the complication-related adverse outcomes for this context.

Based on our impact substudy, occurrence of ECLS and multiple morbidities are particularly important adverse outcomes—they are associated with high mortality, high resource use and lower quality of life for surviving patients over 6 months after surgery. Future routine monitoring and public reporting for these should be considered by centres and the national audit.

In our engagement work, patients' families consistently told us that outcomes affecting the child as a whole over the medium or long term are very important to them. We thus included morbidities such as feeding problems and acute neurological events but note that those types of outcome were the most complicated to capture, and we intend to publish additional material on these topics.

### Limitations

We acknowledge that the definition, measurement, reporting and interpretation of morbidity are much more complex than for mortality. This activity requires significant staff resources and clinician buy in, in particular because some morbidities are challenging to measure and to interpret. The number of individual morbidities was low in some of our analyses, limiting interpretation. We have not captured every single morbidity that exists after children's heart surgery.

### Context

There have been previous attempts at reporting morbidity after paediatric cardiac surgery, however these have in general captured events within a hospital setting, whereas our study captures outcomes out to 6 months postsurgery. A single centre study generated an aggregate 'Morbidity Index' by assigning subjective weights to postoperative complications.[26] The Society of Thoracic Surgery in the USA, developed a 'Morbidity Score', based on data from their multicentre registry.[27] We note that condensing diverse morbidities into a single score may lead to loss of information. Moreover, recent work on using graphical methods to routinely monitor a range of morbidities highlighted the complexity of graphically summarising multiple morbidities.[28] The Pediatric Cardiac Critical Care Consortium (PC⁴) was set up in 2009, and provides partner sites with access to real-time, reliable and actionable data to be used for local quality improvements.[29–32] However, membership of PC⁴ is voluntary for institutions on a subscription basis, the reported measures been selected by clinicians, the reporting of outcomes is accessible only to subscribing member institutions and these are limited to in-hospital measures.

## FUTURE PRACTICE AND RESEARCH

Neurodevelopmental problems are common in children with CHD undergoing surgery[33]; however, we detected acute neurological events (ANE) following only 2.1% of procedures. Therefore, it appears that perioperative 'ANE' represents the tip of the iceberg, and hence the scope of surveillance for child neurodevelopment in CHD needs to extend well beyond the perioperative period.

Despite their great importance to families, we note that participants reported subjectivity in the collection of data on 'feeding problems'; hence, these were difficult to capture consistently. Additional research may help determine the best way to measure this important morbidity and to elucidate approaches to alleviate the impact of feeding problems in CHD.

Our measure of renal failure was the need for renal support, and although relatively easy data to collect this may not be the optimal method to capture this morbidity, given that there are differences in practice between clinicians.[34] Further research may help us to understand the best approach to manage postoperative renal injury in CHD.

Our results perhaps unsurprisingly indicate that parents of children who suffer morbidities experience higher rates of anxiety and depression postoperatively. This emphasises the importance of supporting parents during this phase.

We used study data to co-develop parent and carer information resources, showing what the morbidities are and how their incidence and the length of stay may vary based on the complexity of the child's condition. Parents told us that it helps to know that first, they are not alone in facing a morbidity second, clinical teams have seen morbidities before and know how to deal with them, and third, it is better as a parent to 'be prepared'. Furthermore, parents indicated that information about impact such as nearly all children who experience a morbidity and recover will have a similar quality of life to children who did not experience a morbidity by the 6-month mark, was very helpful to know.

Within the scope of this project, a new Excel tool has been developed and piloted, which enables clinical teams to benchmark and report the local rates of morbidities with a quality assurance goal, such as in a mortality and morbidity conference. Furthermore, we have kept in close contact with the NCHDA and the Clinical Reference Group for CHD services. The NCHDA has already started to collect five of the nine morbidities within the nationally mandated dataset, using the definitions that we developed.

## Author affiliations
[1]Cardiorespiratory Division, Great Ormond Street Hospital for Children, London, UK
[2]Clinical Operational Research Unit, UCL, London, UK
[3]Institute of Child Health, UCL, London, UK
[4]Evelina London Children's Hospital, London, UK
[5]Birmingham Women's and Children's NHS Foundation Trust, Birmingham, UK
[6]Paediatric Intensive Care, Bristol Royal Hospital for Children, Bristol, UK
[7]Health Economics, University College London, London, UK
[8]Congenital Heart Surgery, Royal Hospital for Children, Glasgow, UK
[9]Department of Applied Health Research, University College London, London, UK
[10]Royal Hospital for Children, Glasgow, UK
[11]Bristol Royal Hospital for Children, Bristol, UK
[12]University Hospitals Bristol NHS Foundation Trust, Bristol, UK
[13]Paediatric Intensive Care, Evelina London Children's Hospital, London, UK

**Acknowledgements** The authors would like to thank Martin Utley, Sheryl Snowball, Luke Maidment, Sarah Bohannon, Liz Smith, Kate Penny-Thomas, Joanne Webb, Sinead Cummins, John Stickley, Natasha Khan, Teresa Dickson, Ray Samson, Isobel Mcleod, Paul Wellman, Rhian Lakhani, Kathleen Selway, Carrie Cherrington, Andrew Parry, Rob Tulloh, Bill Gaynor, Rodney Franklin, Lisa Allera, Kate Bull, Trevor Ritchens, Branko Mimic, Jon Smith, Lyvonne Tume, Vibeke Hjortdal, Michael Vath, Tom Treasure, Anne Keatley Clarke, Bea Tuten, and all the patients who participated in the study.

**Collaborators** Cardiac Impact Study Group: Martin Utley, Sheryl Snowball, Luke Maidment, Sarah Bohannon, Liz Smith, Kate Penny-Thomas, Joanne Webb, Sinead Cummins, John Stickley, Natasha Khan, Teresa Dickson, Ray Samson, Isobel Mcleod, Paul Wellman, Rhian Lakhani, Kathleen Selway, Carrie Cherrington, Andrew Parry, Rob Tulloh, Bill Gaynor, Rodney Franklin, Lisa Allera, Kate Bull, Trevor Ritchens, Branko Mimic, Jon Smith, Lyvonne Tume, Vibeke Hjortdal, Michael Vath, Tom Treasure, Anne Keatley Clarke, Bea Tuten.

**Contributors** Designed the study and wrote the protocol: KB, CP, DR, JW, SM and VT. Clinical expertise: VT, DA, DB, SS, AM, KB, SMT, DB, PD and WR. Study implementation, protocol adjustments, data cleaning: KB, AJ, KS, VT, JC, DB, PD, SS, WR and SMT. Statistical analysis: DR, CP, SMT and KB. Health economic analysis: EH and SM. Contributed to the write up and signed off the paper: KB, CP, DR, JW, VT, DA, DB, JC, PD, EH, AJ, AM, SM, WR, KS, SS, SMT and TW.

**Funding** This project was funded by the National Institute for Health Research Health Services and Delivery Research programme (Project No: 12/5005/06). KB, DR, JW and VT were supported by the National Institute for Health Research Biomedical Research Centre at Great Ormond Street Hospital for Children NHS Foundation Trust and University College London.

**Disclaimer** The views and opinions expressed therein are those of the authors and do not necessarily reflect those of the NIHR HS&DR programme or the Department of Health.

**Competing interests** None declared.

**Patient consent for publication** Not required.

**Ethics approval** The study has ethical approval from London City Road Research Ethics Committee (14-LO-1442).

**Provenance and peer review** Not commissioned; externally peer reviewed.

**Data availability statement** For data to be made available specific ethical and HRA approval would be required.

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
