## [Reviewer comments · BMJ Open]

ARTICLE DETAILS

TITLE (PROVISIONAL)	What are the important morbidities associated with paediatric cardiac surgery? : A mixed methods study
AUTHORS	Brown, Katherine; Pagel, Christina; Ridout, Deborah; Wray, Jo; Anderson, David; Barron, David; Cassidy, Jane; Davis, Peter; Hudson, Emma; Jones, Alison; Mclean, Andrew; Morris, Stephen; Rodrigues, Warren; Sheehan, Karen; Stoica, Serban; Tibby, Shane; Witter, Thomas; Tsang, Victor

VERSION 1 – REVIEW

REVIEWER	Nancy Ghanayem Texas Children's Hospital and Baylor College of Medicine. United States of America
REVIEW RETURNED	21-Jan-2019

GENERAL COMMENTS	In this manuscript regarding morbidities associated with paediatric cardiac surgery, the authors address the importance of understanding outcomes beyond mortality. This work is very important to clinicians and caregivers. The manuscript will be an important addition to the literature but warrants some revision based on the following comments and questions:  1. The term “surgical morbidity” implies attribution of outcome to the intraoperative period alone despite the definition embedded within the manuscript. Consider rephrasing to morbidity following surgery or dropping “surgical” from the term since inclusion into the study is surgery. 2. Recruitment of patients without morbidity based on univentricular status seems suboptimal particularly given the heterogenous biventricular population. Why not use procedure or Cardiac Diagnosis Group or even STAT category? 3. Several of the specific morbidities would fall within the Major Adverse Event category. What are the MAEs that are not already mentioned? 4. What is the definition of prolonged pleural effusion? 5. Younger age was a risk factor for morbidity but also is a risk factor for necrotizing enterocolitis. Additionally, neonates and infants who require surgery early and infants are more likely to have more complex disease. How does the analysis adjust for covariates? If it doesn't, further analysis should be done adjusting for age at a minimum. 6. How are pre-operative morbidities accounted for especially those related to feeding or pre-existing neurologic deficits? Are all the captured morbidities postoperative or just present at hospital discharge? 7. Pertinent to comments 5 and 6, an attempt to identify risk factors for morbidity detracts from the objective of this report and is
--

	limited if the morbidity is not time-stamped. The manuscript would serve its purpose and if the results centered on the incidence of morbidity at discharge, what is titled secondary outcomes and the impact of morbidity. Specifically, consider eliminating the section to identify risk factors since there are several limitations to that specific analysis unless there is better matching and modeling based on covariates. 8. The discussion did not take advantage of some important findings. Specifically, there was minimal quality of life and health economic findings. There is an opportunity to speculate on processes or areas of improvement including outpatient complex care coordination to mitigate some of the observations. The emphasis on context overstates the scope of the Pediatric Cardiac Critical Care Consortium (PC4) as it relates to morbidity following discharge. PC4 predominantly captures acquired morbidity while in the intensive care unit and does not capture all hospital acquired morbidity prior to discharge. Future endeavors of PC4 will be to ultimately link ICU and non-ICU hospital outcomes to outcomes following discharge through Cardiac Networks United, a relatively new entity. 9. Finally, the manuscript would have better flow from further editing of as it relates to sentence structure (grammar, use of punctuation), especially the abstract. In closing, this is an important contribution to paediatric cardiac disease, and I look forward to reading a revised manuscript.
--	--

REVIEWER	Dr Michael Waller School of Public Health University of Queensland Australia
REVIEW RETURNED	21-Jan-2019

GENERAL COMMENTS	Thank you for the opportunity to review this paper. I have the following comments:  1. The result from the abstract that among patients with morbidity 6-month survival was 89.2% (85.5, 90.6) does not appear to be stated in the manuscript body. Please can you restate this result in the results section. Also with this statistic please double check the numbers because the CI appears to be uneven (i.e. 89.2 is a lot closer to 90.6 than 85.5). 2. For the bullet points in the abstract be clear what the comparisons are between (e.g. 6 weeks compared to 6 months etc.) 3. Page 5. 1st paragraph. There are 2 commas after "cardiac surgery". Remove one of these. 4. In the Methods section it is important that you clearly define the study design. The terms case and control are mentioned, but I doubt this is a case-control study? A longitudinal cohort study? This is a key omission. 5. In the mixed effects regression model did you account for clustering by pair. Was this the only analysis which used these paired comparisons? Also why were 108 patients unmatched (was this because no match could be found)? Overall I would like the methods to be more clearly articulated. There are at least 3
--

	different statistical models used, so it would be useful to carefully guide the reader through the comparisons being made and the subset of the data being used. Page 13. Table 2. Last row, the percentage listed is 2/0%. I imagine this is a typo. Table 3: This table is very difficult to comprehend. More labels are required and perhaps a more informative title. The information in the table should be able to be understood by the reader, this is not currently the case. It is not clear what is different between rows 2 to 5 and rows 7 to 10. Likewise rows 13 to 15 and 16 to 18. Table 4: The explanation of the results in Table 4 is also difficult to follow. Page 15. Should "Significantly increased (P>0.05)" be "(P<0.05)" instead? Also the wording used for the events listed are different to those presented in Table 4. e.g. Is "un-planned re-intervention" the same as "unplanned reoperation" or something different? Typo on page 17. "Despite their great important" change to "importance"
--	---

VERSION 1 – AUTHOR RESPONSE

In this manuscript regarding morbidities associated with paediatric cardiac surgery, the authors address the importance of understanding outcomes beyond mortality. This work is very important to clinicians and caregivers. The manuscript will be an important addition to the literature but warrants some revision based on the following comments and questions:

1. The term “surgical morbidity” implies attribution of outcome to the intraoperative period alone despite the definition embedded within the manuscript. Consider rephrasing to morbidity following surgery or dropping “surgical” from the term since inclusion into the study is surgery.

Thanks we have dropped surgical from the term throughout the paper.

2. Recruitment of patients without morbidity based on univentricular status seems suboptimal particularly given the heterogenous biventricular population. Why not use procedure or Cardiac Diagnosis Group or even STAT category?

The selection of matching criteria was challenging and we had to trade-off between what might be considered a perfect match and considerations of feasibility. Our first choice was to match on age group and RACHS category plus single / biventricular status. We prospectively set out in the study protocol that if all three criteria proved to be infeasible we would drop the requirement for a RACHS category match.

One issue to consider is that given inter centre variability in certain aspects of care and treatment we were consistently advised to match within centres. Matching within centre plus the finite timeline of the study meant that we had to use reasonably broad categories. Attempting to match based on individual diagnosis or procedure would have led to a majority of patients being unmatched. In terms of broad groups, STAT categories are unfamiliar in the UK as they are not used and clinical teams are more familiar with RACHS. In the first 9 months of the study we matched on age group and RACHS category plus single / biventricular status, however at that stage we were advised by the steering committee to drop the RACHS category requirement for matching, because this was leading to an excessive rate of unmatched patients.

This issue is reported in detail in our grant report which covers the whole study in full and will be published in NIHR Journals Library.

3. Several of the specific morbidities would fall within the Major Adverse Event category. What are the MAEs that are not already mentioned?

Given the reviewer's queries about definitions, we have added an appendix detailing the morbidity definitions, drawn from our paper Brown KL, Pagel C, Brimmell R, et al. Definition of important early morbidities related to paediatric cardiac surgery. *Cardiol Young*. 2017;27:747-56 doi: 10.1017/S1047951116001256 [published Online First: 2016/09/30].

MAE includes:

- Cardiac arrest, where the child receives any chest compressions or defibrillation.
- Chest re-opening on the ICU or ward for any reason.
- Major haemorrhage in the ICU following surgery.
- A 'Never Event' applicable to paediatric cardiac surgery as selected from the 'Never Events' list published for NHS for 2015

(wrong site or wrong patient surgery, wrong prosthesis surgery, retained foreign object post procedure, wrong route administration of medication, transfusion or transplantation of main red cell group incompatible blood components or organs, misplaced naso-gastric or oro-gastric tubes, Tissue injury to limb or vital organ such as perforated viscus or ischaemic limb injury.)

4. What is the definition of prolonged pleural effusion?

Either a chylous pleural effusion or significant chylous pericardial effusion or significant chylous ascites or a prolonged non-chylous effusion that necessitates thoracic drainage at least ten days following index cardiac surgery.

5. Younger age was a risk factor for morbidity but also is a risk factor for necrotizing enterocolitis. Additionally, neonates and infants who require surgery early and infants are more likely to have more complex disease. How does the analysis adjust for covariates? If it doesn't, further analysis should be done adjusting for age at a minimum.

We have re structured the statistical methods section as this was not clear enough, and we hope that our approach to the adjustment of covariates is now more evident to the reader. (see page 10)

The results of the multiple logistic regression model for 'any morbidity' presented in Table 1 are expressed as odds ratio for 'any morbidity' by the stated factor, adjusted for age band, low weight, cardiac diagnostic category, functionally univentricular heart, acquired comorbidity, congenital comorbidity, severity of illness indicators, additional cardiac risk factors, specific procedure group, bypass time category. Therefore the multivariable logistic regression model (Table 1) includes all significant independent risk factors for morbidity, including age. We have not considered risk factors for individual morbidities in this manuscript because there are too few cases. Future descriptive work will explore this in greater detail.

Where we have studied the impact of morbidities with regards to quality of life and costs we have adjusted all models for important covariates and we have highlighted this in the manuscript more clearly listing these co variates within the description of Tables 3 and 4.

6. How are pre-operative morbidities accounted for especially those related to feeding or pre-existing neurologic deficits? Are all the captured morbidities postoperative or just present at hospital discharge?

This issue is addressed for each morbidity individually in the new Appendix drawn from our paper Brown KL, Pagel C, Brimmell R, et al. Definition of important early morbidities related to paediatric cardiac surgery. *Cardiol Young*. 2017;27:747-56 doi: 10.1017/S1047951116001256 [published Online First: 2016/09/30].

For acute neurological event: Includes neurological morbidities that, based on best clinical judgement, arose as new findings around the time of surgery that were detected within the same hospitalisation as the surgery. It is recognised that in certain circumstances such as where a child is very sick on life support, pre procedure assessment is challenging, in these circumstances as full an evaluation as possible to be completed, incorporating serial assessments over time.

For feeding problems: A diagnosis of post-operative feeding problems should be considered during recovery after surgery and prior to discharge from the specialist centre either to home or to secondary care if the child is unable to feed normally. The goal is detection of feeding problems which are new post-surgery, and it is recognised that this may be challenging where a child was not fed pre-operatively for cardiac reasons since feeding ability will not have been assessed objectively.

7. Pertinent to comments 5 and 6, an attempt to identify risk factors for morbidity detracts from the objective of this report and is limited if the morbidity is not time-stamped. The manuscript would serve its purpose and if the results centered on the incidence of morbidity at discharge, what is titled secondary outcomes and the impact of morbidity. Specifically, consider eliminating the section to identify risk factors since there are several limitations to that specific analysis unless there is better matching and modeling based on covariates.

We have edited methods to clarify as follows:

Morbidities were attributed to the immediately preceding cardiac surgery and defined within the same hospitalisation, except for unplanned re operation which was defined as an unanticipated procedure within 30-days and mediastinitis which could be identified post-discharge by the operating surgeon (part of post-surgical infection morbidity see Appendix and reference 8 for details).

The definitions appendix provides additional clarity as to the time stamp of each morbidity including for example specific exclusion criteria based on presence of a morbidity before surgery). Data collection on the occurrence of morbidities was prospective and based on consistently applied definitions. We hope that the revised account makes our analyses of morbidity risk factors clearer, we think this is an important part of the account.

8. The discussion did not take advantage of some important findings. Specifically, there was minimal quality of life and health economic findings. There is an opportunity to speculate on processes or areas of improvement including outpatient complex care coordination to mitigate some of the observations. The emphasis on context overstates the scope of the Pediatric Cardiac Critical Care Consortium (PC4) as it relates to morbidity following discharge. PC4 predominantly captures acquired morbidity while in the intensive care unit and does not capture all hospital acquired morbidity prior to discharge. Future endeavors of PC4 will be to ultimately link ICU and non-ICU hospital outcomes to outcomes following discharge through Cardiac Networks United, a relatively new entity.

Thanks for the suggestion, we have edited the discussion along these lines in respect to PC4 and have expanded our comments on the impact outcomes.

9. Finally, the manuscript would have better flow from further editing of as it relates to sentence structure (grammar, use of punctuation), especially the abstract.

Thanks we have edited all sections of the paper trying to achieve better flow and sentence structure.

In closing, this is an important contribution to paediatric cardiac disease, and I look forward to reading a revised manuscript.

Reviewer: 2

Reviewer Name: Dr Michael Waller

Institution and Country: School of Public Health University of Queensland Australia Please state any competing interests or state 'None declared': None declared

Please leave your comments for the authors below Thank you for the opportunity to review this paper. I have the following comments:

1. The result from the abstract that among patients with morbidity 6-month survival was 89.2% (85.5, 90.6) does not appear to be stated in the manuscript body. Please can you restate this result in the results section. Also with this statistic please double check the numbers because the CI appears to be uneven (i.e. 89.2 is a lot closer to 90.6 than 85.5).

Thanks for pointing it out, the CIS were asymmetric however we have corrected the error that was picked up and we added this data to Table 2 in the results section.

2. For the bullet points in the abstract be clear what the comparisons are between (e.g. 6 weeks compared to 6 months etc.)

Good point we have done this.

- Morbidity related impairment in health related quality of life improved between 6 weeks and 6 months.
- There were a median of 13 (95% CI (10.2, 15.8) $P < 0.001$) fewer days at home with morbidity compare to no morbidity.

3. Page 5. 1st paragraph. There are 2 commas after "cardiac surgery". Remove one of these.

Done

4. In the Methods section it is important that you clearly define the study design. The terms case and control are mentioned, but I doubt this is a case-control study? A longitudinal cohort study? This is a key omission.

We have separated the headers of design and setting.

The design was a prospective, multi-centre, multidisciplinary mixed methods study. Selection of morbidities was based on consensus methods. Morbidity incidence was evaluated with a prospective cross sectional study. Measurement of morbidity impact entailed a prospective, case-matched, longitudinal study.

5. In the mixed effects regression model did you account for clustering by pair. Was this the only analysis which used these paired comparisons? Also why were 108 patients unmatched (was this because no match could be found)? Overall I would like the methods to be more clearly articulated. There are at least 3 different statistical models used, so it would be useful to carefully guide the reader through the comparisons being made and the subset of the data being used.

We have restructured the methods section to make it clearer in response to feedback. There is now a single section covering all of the statistical and health economic data analysis. The first two statistical models (mixed effects regression and quantile regression) were adjusted for clustering by pair.

There were 108 unmatched patients, where no match could be found. Please see the response to reviewer 1 point 2 about why this was and the process followed.

Page 13. Table 2. Last row, the percentage listed is 2/0%. I imagine this is a typo.

Corrected.

Table 3: This table is very difficult to comprehend. More labels are required and perhaps a more informative title. The information in the table should be able to be understood by the reader, this is not currently the case. It is not clear what is different between rows 2 to 5 and rows 7 to 10. Likewise rows 13 to 15 and 16 to 18.

Table 3 and its title / description has been changed to make it clearer.

Table 4: The explanation of the results in Table 4 is also difficult to follow. Page 15. Should "Significantly increased ($P>0.05$)" be " $(P<0.05)$ " instead? Also the wording used for the events listed are different to those presented in Table 4. e.g. Is "un-planned re-intervention" the same as "unplanned reoperation" or something different?

Thanks we have corrected the typo and have edited this section to make it clearer including ensuring all the terms are congruent with the rest of the paper.

Typo on page 17. "Despite their great important" change to "importance"
This has been corrected.

VERSION 2 – REVIEW

REVIEWER	Nancy Ghanayem Baylor College of Medicine Texas Children's Hospital United States
REVIEW RETURNED	11-Jun-2019

GENERAL COMMENTS	The revisions have been satisfactorily addressed. The manuscript reads well and will be an important contribution to the literature.
--

REVIEWER	Michael Waller University of Queensland, School Public Health
REVIEW RETURNED	23-May-2019

GENERAL COMMENTS	I am satisfied with the responses to my original comments.
--